# FlexTab: Towards a Flexible Encoder-Decoder Architecture for Tabular In-Context Learning

**Marek Polewczyk** [* 1]  **Maximilian Schambach** [* 1]  **Marco Spinaci** [* 2]  **Sam Thelin** [1]  **Johannes Höhne** [1]

## Abstract

We introduce FlexTab, a flexible encoder-decoder architecture for in-context learning on tabular data that combines a single, task-agnostic encoder with a suite of task-specific decoders. Unlike existing tabular in-context learners, which entangle feature representations with a specific prediction target, our design produces *target-agnostic* row embeddings that can be leveraged across a range of downstream tasks. We demonstrate this flexibility on three distinct tasks: classification, regression, and entity matching. We note that our architecture can be extended to problems such as outlier detection, clustering, and entity classification in relational databases, among others. Both the encoder and the task-specific decoders are trained on a large corpus of real-world, unlabeled tables. FlexTab achieves state-of-the-art or competitive performance across the examined tasks. Our results demonstrate that a single shared encoder, paired with task-specific decoders, can serve as an effective general-purpose backbone for diverse tabular prediction problems.

## 1. Introduction

Recently, tabular in-context learners such as TabPFN, TabICL, and ConTextTab (Hollmann et al., 2025; Qu et al., 2025; Spinaci et al., 2025) have been shown to outperform conventional per-dataset trained and tuned models across a range of tabular prediction tasks. Their performance is impressive, outperforming established baselines such as gradient-boosted trees (Chen & Guestrin, 2016; Prokhorenkova et al., 2018; Ke et al., 2017), as well as recent state-of-the-art (SOTA) deep-learning approaches (Holzmüller et al., 2024; Gorishniy et al., 2025), and even automated machine learning solutions such as Au-

toGluon (Erickson et al., 2020). Tabular in-context learners open up many new possibilities for applications in both research and industry. However, their development and application have mostly focused on standard supervised prediction tasks, namely classification and regression. While some variations exist, in particular in the TabPFN ecosystem (e.g., providing embeddings, interpretability, outlier detection, missing value imputation, etc.) as well as problem-specific adaptations (e.g., causal inference, time-series forecasting, or graph predictions), these are mostly created in a post-hoc fashion, by recasting the problem as a single-table classification or regression task, or require significant architectural adaptations and full retraining.

While effective in many cases, this general approach exhibits the following limitations: (a) the necessity to reformulate every (sub)problem as a single-table classification or regression task; (b) the tight coupling between internal latent representations and predictive modeling, which causes features to always be conditioned on targets; and (c) synergies between different tasks are not leveraged.

We argue that these limitations cannot be overcome within the design principles of current tabular in-context learners, due to the inherent entanglement of internal target-aware representations and predictive decoding. To this end, we propose a *flexible* tabular in-context learning architecture, FlexTab, based on an encoder-decoder setup that clearly separates internal latent representations from predictive paths by using a variety of problem-specific decoders. We train our approach using a large collection of real-world (unlabeled) tables, and evaluate its effectiveness on well-studied classification and regression cases, as well as on entity matching, highlighting its flexibility across single- and multi-table scenarios. We match or surpass previous SOTA performance across all investigated scenarios.

## 2. Tabular In-Context Learning

### 2.1. Background

Tabular in-context learners were primarily pioneered by work on prior-fitted networks and (conditional) neural processes (Müller et al., 2022; Garnelo et al., 2018), resulting in the first version of TabPFN (Hollmann et al., 2023). Build-

---

[*]Equal contribution  [1]SAP SE, Germany. [2]SAP France. Correspondence to: Marek Polewczyk <marek.polewczyk@sap.com>.

*Proceedings of the $2^{nd}$ ICML Workshop on Foundation Models for Structured Data*, Seoul, South Korea. 2026. Copyright 2026 by the authors.

*Figure 1.* A simplified schematic overview of our proposed architecture, separating an in-context encoder from task-specific decoders. Note that details such as normalization and feed-forward layers, as well as skip connections, are omitted.

ing on its core in-context learning principles, a multitude of works extended and refined TabPFNv1, ranging from changes in the synthetic pretraining priors (Qu et al., 2025; Zhang et al., 2025), to incorporating retrieval (Ma et al., 2024), adapting the attention mechanism from row-wise token attention to cell-wise alternating 2D-attention (Hollmann et al., 2025), improving accuracy and speed (Qu et al., 2025; 2026), and handling semantic features (Spinaci et al., 2025), to name a few. Most of the prior art focuses on tabular classification and regression tasks. While some works exist that transfer or apply tabular in-context learners to non-tabular domains such as time series forecasting (Hoo et al., 2024), reinforcement learning (Schiff et al., 2025), or the graph domain (Hayler et al., 2025; Eremeev et al., 2025), applications to other tabular tasks remain scarce. Several works have investigated causal inference (Robertson et al., 2025; Balazadeh et al., 2025), unsupervised clustering (Marszałek et al., 2025; Zhao et al., 2026), and anomaly detection (Marszałek et al., 2026; Wei & Armanfard, 2026), including very recent preprints. However, these approaches require fundamental changes to the model architecture and/or the pretraining data. We argue that tabular foundation models should natively handle and generalize to a multitude of tabular tasks without the need to recast problems as single-table classification or regression tasks, and without substantial architectural adaptations.

While some additional applications are supported in community-maintained extensions of TabPFN (e.g., outlier detection) and some may be approached using embeddings extracted from current models, these approaches have not been experimentally investigated and come with their respective shortcomings. For example, embeddings from current models such as TabPFN, TabICL, and ConTextTab are *target-aware*. Obtaining embeddings from rows or tables without a target (e.g., for clustering or matching) requires dummy targets, e.g., by adding random ones or treating features as targets, which hurts performance and/or runtime.

## 2.2. Proposed architecture

Our proposed architecture is depicted in Figure 1 in a simplified way, with more details presented in Appendix A. At its core, our architecture separates general (target-agnostic) context and query row embeddings from task-specific decoding. The target decoding is performed by separate task-specific decoders that contextualize the targets with the feature row embeddings via cross-attention. The encoder and decoders are built around 2D-attention blocks, with alternating cross-row (vertical) and cross-column (horizontal) attention mechanisms – analogous to the design of TabPFN, Mitra, and ConTextTab – but separate the target prediction stream into different decoders to make the feature encodings target-agnostic and thus usable across a range of problems.

**Encoder:** The encoder model builds on the general setup of the ConTextTab architecture (Spinaci et al., 2025), adopting the same embedding strategy with separate numerical, categorical, and time embedding layers. Given input tables with batch size $B$, $R$ rows, and $C$ columns, the data is embedded into a tensor of shape $(B, R, C, D)$, where $D$ denotes the hidden dimension. A learnable [ROW] token (replicated across rows) is appended to the embeddings along the column axis, resulting in a tensor of shape $(B, R, C + 1, D)$, similar to [CLS] tokens in BERT-like models (Devlin et al., 2019; Liu et al., 2019) and TabICL (Qu et al., 2025; 2026), where these tokens are used to generate row-wise (but target-aware) embeddings. After embedding, cell tokens and [ROW] tokens are jointly contextualized through alternating cross-column and cross-row attention layers. When the input table consists of both context and query rows, cross-row attention is performed via cross-attention in such a way that context tokens only attend to other context tokens, and query tokens do not attend to each other. Similarly, cross-column attention is structured so that [ROW] tokens attend to feature tokens, while feature tokens cannot attend back.

Rather than relying only on the final-layer representations, we collect the [ROW] token outputs from every encoder layer. Each layer's representation is passed through a dedicated linear projection, after which the resulting representations are summed across layers. The aggregated representation is then projected by a shared linear layer to produce the final encoder output. This design is markedly different from conventional encoder-decoder designs, where typically the last encoder layer's output is used (Vaswani et al., 2017).

In our design, the encoder excludes any target columns and instead uses trainable [ROW] tokens, resulting in general-purpose, target-agnostic row representations of shape $(B, R, D)$ that are used as the cross-attention $KV$ inputs to the task-specific decoders.

**Decoders:** The different decoders consist of the same 2D-attention blocks as the encoder, with the crucial difference

that cross-column attention is realized via cross-attention to the encoder outputs. As in common tabular ICL architectures, query targets are decoded from a learnable `[MASK]` token. The key distinction is that cross-row attention is applied exclusively to the target column, while the encoder's `[ROW]` tokens are provided as attention $KV$ inputs to all cross-column attention layers, with the target column acting as the attention $Q$. In this way, target tokens attend to the separately encoded features via cross-attention. In the main classification and regression decoder, each target attends to a single row embedding. However, in the decoder for matching, the target attends to two tokens that originate from separate tables with potentially different schemas. While not the focus of this initial investigation, our architecture also naturally extends to incorporate other external embeddings, enabling the decoder to integrate heterogeneous data sources beyond tabular inputs.

**Classification and regression:** The main design follows the description above, with separate heads for regression and classification but otherwise shared decoder weights. The classification head is identical to that used in ConTextTab. However, unlike ConTextTab, we treat regression as classification and predict logits over quantile bins instead of producing a direct point estimate. In the default setting, we match the number of transformer blocks $N$ to that of the encoder. Additional details on the target embedding and prediction heads are provided in Appendix A.

**Matching:** We consider cross-table row matching as a pairwise binary classification task. That is, row pairs are treated as matching targets, and the two tables are assumed to have the same number of rows. In particular, we do not consider the steps of a full matching pipeline such as blocking (Konda et al., 2016) for creating these pairwise candidate tables, but instead focus on the prediction step. To apply a tabular in-context learning model to pairwise cross-table matching, one could recast the task as a single-table problem by concatenating the rows into one table. However, we propose to first encode each table separately, using only the rows of the respective table as context, and then decode the binary matching target using cross-attention to the two separate row embeddings in the decoder stream, as depicted in Figure 1. The binary classification head is architecturally identical to the default classification head, but with the restriction to two target classes.

## 3. Training Details and Task Generation

**Pretraining:** First, we pretrain the encoder together with the classification and regression decoder on conventional tabular prediction tasks, similar to previous tabular in-context learners. We then freeze the encoder and train the additional matching decoder on top. We do this because the generation of classification and regression tasks for large-

scale tabular in-context learners is the most well-studied and has proven effective, including when using real-world data (Ma et al., 2024; Spinaci et al., 2025; Ma et al., 2025).

Throughout, we use a large corpus of ca. 300 k unlabeled tables from an internal, proprietary dataset compiled from different public sources. By default, we use 12 layers and a hidden dimension of 768, with 12 heads of 64 dimensions, and a feed-forward dimension of 3072, for a total of ca. 175 M encoder parameters.

**Classification and Regression:** We generate classification and regression tasks from randomly sampled real-world tables in a similar fashion to prior work (Spinaci et al., 2025). At each step, we randomly select 1000 rows, then sample between 50 and 900 rows as queries, and use the rest as context. Subsequently, we randomly sample feature and target columns. We train the model on 8 H100 GPUs for 30 M steps until convergence, using a micro-batch size of 1 and accumulating gradients to an effective batch size of 256. Throughout, we use the AdamW optimizer with a learning rate of 1E−4. For both classification and regression, we use a cross-entropy loss.

**Matching:** Using the frozen, pretrained encoder, we train the matching decoder on matching tasks generated from real-world tables. To this end, we use two equally weighted strategies: a same-row matching and a hidden target-based matching construction. For the former, we create synthetic matching pairs by splitting a randomly drawn table into two tables with a random overlap, and then shuffle the tables. Pairs that originate from the same original row are considered matching, others as non-matching. For the second approach, we follow the same procedure but consider rows matching if they agree on a held-out target. We describe this sampling procedure in more detail in Appendix A. The training loss is a cross-entropy loss on the predicted binary targets. The decoder is trained for ca. 12 M steps.

## 4. Results

**Classification and Regression:** We evaluate on a mix of conventional, numeric-heavy benchmarks as well as semantic ones. For the conventional benchmarks, we evaluate on TabArena-Lite (Erickson et al., 2025) and TALENT-Tiny (Ye et al., 2024), and for the semantic ones, we use CARTE (Kim et al., 2024) and TextTab (Mráz et al., 2025). Throughout, we report mean accuracy and soft-clipped $R^2$, as well as mean rank. We compare against a range of best-in-class baselines, including SOTA in-context learners as well as per-dataset trained models, tuned and ensembled via 5-fold inner cross-validation. We provide additional evaluations and baseline details in Appendices A.5 and B.1.

The main results are shown in Table 1. FlexTab performs competitively across non-semantic benchmarks while set-

*Table 1.* Classification and regression performance, depicting mean accuracy (Acc) for classification and (soft-clipped) $R^2$ score for regression tasks, in percent, as well as mean per-task rank.

| Model | All | CARTE | | | TabArena-Lite | | | TALENT-Tiny | | | TextTab | | |
|---|---|---|---|---|---|---|---|---|---|---|---|---|---|
| | Rank | Rank | Acc | R2 | Rank | Acc | R2 | Rank | Acc | R2 | Rank | Acc | R2 |
| **FlexTab (ours)** | **2.6** | **1.5** | **77.3** | **74.4** | 3.3 | 87.1 | 79.5 | 3.1 | 88.0 | **85.8** | **2.5** | 83.8 | 62.7 |
| RealMLP | 3.1 | 3.4 | 74.3 | 69.0 | 2.9 | 88.3 | 79.9 | 2.9 | **88.6** | 85.0 | 3.2 | 82.3 | **67.7** |
| TabPFN v2.6 | 3.1 | 4.9 | 70.7 | 59.9 | **1.7** | 88.7 | **80.5** | 2.2 | 88.4 | 84.9 | 3.8 | 81.5 | 63.8 |
| ConTextTab | 3.2 | 2.4 | 77.1 | 72.4 | 3.6 | 87.6 | 77.8 | 3.7 | 87.6 | 83.2 | 3.0 | **84.4** | 58.8 |
| TabICL v2 | 3.2 | 5.2 | 70.5 | 55.3 | 2.0 | **88.7** | 79.9 | **2.1** | 88.3 | 85.3 | 3.5 | 82.5 | 60.2 |
| CatBoost | 3.4 | 3.4 | 76.3 | 68.3 | 3.4 | 88.2 | 79.0 | 3.7 | 87.1 | 83.5 | 2.8 | 83.7 | 65.4 |

*Table 2.* Matching benchmark F1 scores (in %, positive class) obtained as 5-fold cross-validation averages for each dataset. The average per-group rank is denoted as GRk whereas the average rank across all investigated models is denoted as Rk.

| Model | Benchmark | | | | | Average | | |
|---|---|---|---|---|---|---|---|---|
| | Febrl4 | F.-Zagats | Bikes | eBooks | Movies | F1 | GRk | Rk |
| *In-context learners* | | | | | | | | |
| TabPFN | 97.3 | 29.4 | 81.8 | 67.9 | 0.0 | 55.3 | 3.2 | 5.2 |
| TabICL | 95.0 | 36.9 | 82.4 | 73.1 | 0.0 | 57.5 | 2.8 | 4.8 |
| ConTextTab | 83.2 | 23.2 | 82.3 | 60.3 | 79.0 | 65.6 | 3.6 | 5.4 |
| **FlexTab (ours)** | 79.8 | 12.7 | 84.3 | 72.6 | 61.0 | 62.1 | 3.6 | 5.4 |
| **FlexTab-Match (ours)** | **99.3** | **92.5** | **86.8** | **87.3** | **90.2** | **91.2** | **1.0** | **2.4** |
| *Per-dataset trained* | | | | | | | | |
| Deepmatch Hybrid | 99.8 | 87.1 | 87.7 | 88.5 | **87.6** | **90.2** | 1.6 | 2.0 |
| DITTO | **100.0** | **94.7** | **95.5** | **89.5** | 44.9 | 84.9 | **1.2** | **1.8** |

ting a new standard on the semantic CARTE and TextTab benchmarks, highlighting the effectiveness of our approach. Compared to TabPFN and ConTextTab, these results show that target-agnostic row embeddings can achieve SOTA results. We note that this difference is statistically significant in the case of semantic-heavy benchmarks, as shown in the critical-difference diagrams in Appendix B.1.

**Matching:** For benchmarking, we use a total of 5 matching datasets from prior work. However, we note that most public matching datasets cannot be considered tabular but are rather free-text NLP entity matching tasks. Hence, we focus on tabular matching tasks with at least 2 columns of relevant information. In particular, we evaluate on the synthetic Febrl4 dataset (Christen, 2008), as well as Fodors-Zagats, Bikes, eBooks, and Movies, sourced from the DeepMatcher study (Mudgal et al., 2018) as well as the Magellan Data Repository (Das et al.). We perform 5-fold cross-validation and report average match-class F1 scores. We compare our approach against tabular in-context learners by recasting the cross-table matching task as a single-table binary classification problem, as well as against native matching baselines that are trained per dataset: here, we report the Hybrid Deep-Matcher variant (Mudgal et al., 2018) as well as DITTO (Li et al., 2020), but show additional ones in Appendix B.2.

The results are depicted in Table 2. We observe that all tabular in-context learners perform rather poorly. While TabPFN and TabICL seem to perform reasonably well on the syn-

thetic exact-match Febrl4 task, their performance degrades on more semantically heavy tasks, such as Fodors-Zagats or Movies. ConTextTab and FlexTab, which natively handle semantic columns, show slightly more robust performance, but overall fall behind SOTA matching models. With a dedicated matching decoder trained on matching tasks, FlexTab-Match performs very strongly. Our in-context approach even outperforms dedicated per-dataset trained deep learning baselines such as those from the DeepMatcher study or DITTO in terms of average match-class F1. We provide additional results in Appendix B.2.

## 5. Conclusion

**Limitations:** Despite the encouraging results, our approach has the following limitations. First, compressing each row into a single embedding may introduce an information bottleneck, particularly for wide tables or tasks requiring fine-grained cell-level reasoning. Second, as with other models trained on large-scale real-world collections, we cannot fully rule out data contamination between training and evaluation.

**Outlook:** Our work highlights the general efficacy of target-agnostic row embeddings coupled with task-specific decoders. Here, we investigate classification, regression, and matching, but our architecture naturally extends to a range of tabular predictive problems, as well as to the incorporation of non-tabular knowledge sources.

## Acknowledgements

We would like to thank Johannes Hoffart and Markus Kohler for their insightful comments and suggestions throughout the development of this work.

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

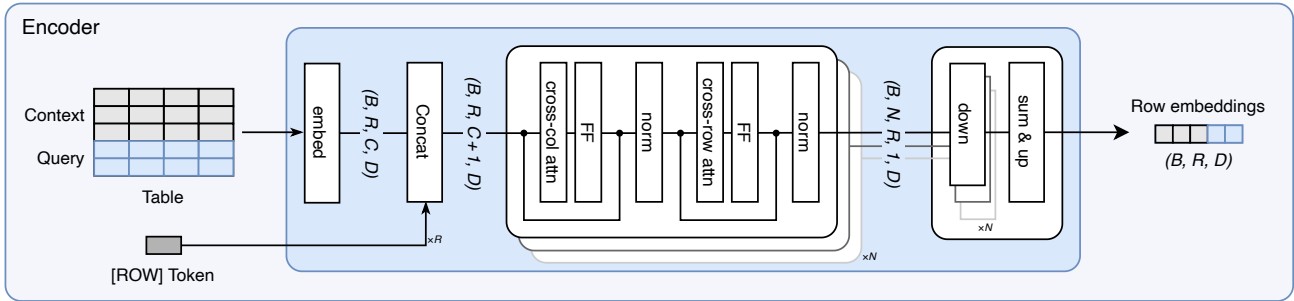

*Figure 2.* Detailed schematic depiction of the encoder architecture. Note that cross-row attention is performed in a way such that query rows cannot attend to each other, analogous to existing architectures. The learnable [ROW] tokens are repeated along the row axis and then appended to the table embeddings along the column axis. Although depicted in parallel, the transformer blocks are executed sequentially, using the output of the previous layer as the input to the next, as is common. However, the $N$ individual [ROW] token outputs of each transformer block are separately (down-)projected, then summed and (up-)projected to obtain the final row embeddings.

# A. Architectures and Pretraining Details

## A.1. Encoder

A more detailed depiction of our proposed encoder is shown in Figure 2. While depicted in parallel, the 2D Transformer blocks are executed sequentially. The visualization further illustrates that each block's output is separately (down-)projected, then summed and (up-)projected to obtain the final row embeddings.

In all cases, cross-row attention is performed in such a way that context rows can only attend to each other and query rows can only attend to context rows, avoiding any query row leakage. This is achieved efficiently using a cross-attention formulation. Given an input sequence of shape $(B, S, D)$ with a batch size $B$, sequence length $S$, and hidden dimension $D$, where the first $C$ elements in the sequence are context examples, and the last $Q$ elements are query examples, we achieve the desired attention pattern by using the context-truncated sequence of shape $(B, C, D)$ as attention $KV$ and the full sequence of shape $(B, S, D) = (B, C + Q, D)$ as the attention $Q$. If no context/query split is present, for example, when encoding a table for matching, all sequence elements are effectively treated as context, and the formulation falls back to full self-attention over all rows.

All decoders are built around the same 2D attention block, using intermediate feed-forward and normalization layers as depicted in Figure 2.

## A.2. Classification and regression

In this section, we describe the decoder architecture used for both classification and regression tasks. The decoder consists of the main backbone and two separate MLP-based heads: one dedicated to classification and the other to regression.

**Classification:** For classification, we employ a standard cross-entropy loss applied to the output of an MLP with a fixed output dimension of 64, corresponding to the maximum number of supported classes. As a result, during inference, the model cannot predict more than this predefined number of classes without relying on less efficient approaches such as hierarchical classification (e.g., as used in TabPFN's many-class extension or TabICL). To ensure the model can associate semantic meaning with class indices (e.g., class "0"), it must learn an internal representation of these identifiers. To facilitate this, we introduce a dedicated input encoding specifically for the target column, in addition to the encodings used for feature columns. Although this design choice breaks class order permutation equivariance, we find it to be effective in typical few-class classification settings.

**Regression:** We encode numerical values via piecewise linear interpolation over a set of learned reference vectors. Specifically, given the training data (context rows), we compute $n{=}50$ quantiles at orders $\frac{1}{2n}, \frac{3}{2n}, \dots, \frac{2n-1}{2n}$ and extend them with a (clipped) minimum and maximum, yielding $n{+}2$ reference points $q_0 < q_1 < \cdots < q_{n+1}$. Each reference point $q_i$ is associated with a learnable vector $v_i$. A value $x \in [q_i, q_{i+1}]$ is then encoded as $(1{-}\lambda)\, v_i + \lambda\, v_{i+1}$, where $\lambda = \frac{x - q_i}{q_{i+1} - q_i}$; values outside $[q_0, q_{n+1}]$ are clipped to the boundary.

For the regression target, this encoding is used on the input side during training. On the output side, regression is cast as

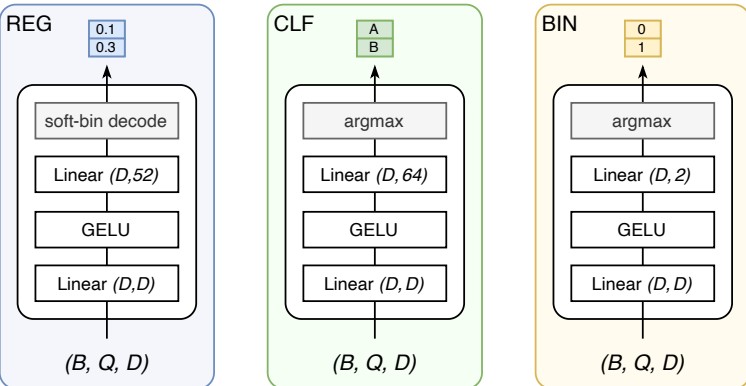

*Figure 3.* Details of the heads used in the different decoders, including the regression (REG) head, classification (CLF) head, and binary classification (BIN) matching head. Grey blocks denote inference-only blocks, whereas for training, the raw logit outputs are used.

classification over the $n+2$ reference points: the model produces a probability distribution $(p_0, \ldots, p_{n+1})$ via a softmax, and the final prediction is the expectation $\sum_i p_i\, q_i$.

### A.3. Matching

For the matching decoder training, we employ two separate pretraining and task generation strategies:

First, from a random table, we generate two column-wise splits with a random overlap of *at most* 50%. These act as the matching Table A and Table B depicted in Figure 1. With a random probability between 5% and 33% (drawn and fixed for the current table), we then mark each row pair as a match by drawing from the Bernoulli distribution. This marking serves as the binary target label of the current sample. We then keep the order of rows marked as matched in place, while randomly shuffling the remaining ones. This strategy resembles cases of pairwise row matches with partially identical column entries. That is, in most cases, the prediction can be made by exact matching, $\hat{y} = (A_i == B_j)$ for some column $A_i$ in Table A and $B_j$ in Table B, respectively. However, note that this cannot trivially be exploited in our architecture as both rows are first encoded into latent [ROW] tokens, which are contextualized over separate statistics and table schemas. Hence, we found this strategy to provide reasonably challenging matching tasks that closely reflect many real-world matching scenarios involving exact or fuzzy column matches.

As for the second strategy, we use a more indirect semantic matching target: Again, using a randomly drawn table, we use the same strategy as in classification training to generate a feature-target split $X, \tilde{y}$ of a corresponding classification target. We then again create two column-wise splits from the feature matrix $X$ with *at least* 50% overlap. The two obtained tables are then shuffled. The ground truth matching target is then defined as the agreement of the original classification target between the rows in Tables A and B, $y = (\tilde{y}_A == \tilde{y}_B)$. That is, two rows are considered matching if they share their underlying classification target. This results in a much more indirect, semantic matching task as opposed to the direct column matching above.

### A.4. Pretraining Data

Our pretraining corpus consists of roughly 300 k raw, unlabeled tables. These tables were sourced and curated at scale from various public sources with a permissive license. An overview of the size distribution is presented in Figure 4, which compares the row, column, and data type distributions of our corpus with those of The Tremendous TabLib Trawl (T4) collection (Gardner et al., 2024). We observe that our pretraining corpus is more broadly spread, particularly in terms of table width and height, and overall has a higher average proportion of string columns. Note that these data type rates are averages of per-table rates. That is, we first measure the data type rate across columns of each table and then collect averages across tables.

Note that we do not perform any contamination study of the corpus with respect to the evaluation benchmarks used, due to the scale and fuzzy nature of the problem and the lack of research on achieving robust results at scale. However, in practice we believe the impact to be small or negligible. Each table is sampled only a few times during training and subsequently augmented, subsampled, and used with a random prediction target. Given the comparatively small model size and in-context

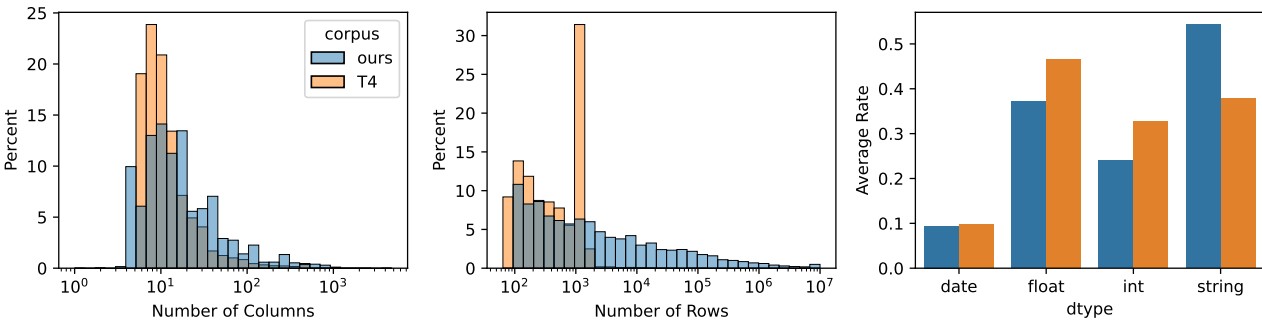

*Figure 4.* Column and row count distribution, as well as per-column data type rate of our pretraining data corpus and the T4 corpus. Our corpus consists of roughly 300 k tables in total, sourced from various public sources. The T4 corpus was downsampled to 500 k tables for its statistics calculation.

learning training objective, we believe that current tabular in-context learners, including our proposed architecture, do not suffer or gain from in-weight memorization of data in any meaningful way. However, at-scale detection of table duplication (e.g., to be used for a contamination study) is an interesting direction for future research.

### A.5. Baselines

Overall, we follow the evaluation protocol and baseline implementations of ConTextTab (Spinaci et al., 2025), but with updated baseline versions:

**TabPFN:** We evaluate TabPFN v2.6 using the official PyPI package version 7.1.1. For classification tasks with more than 10 classes, we use the many-class extension from `tabpfn-extensions` in version 0.2.2+ at the latest git commit `452708bdf2e644dd1b2ebe938862a9ea52ac7e21`, which adds compatibility with TabPFN v2.6. We use the default initialization parameters, in particular using 8 ensembled estimators for inference.

**TabICL:** We evaluate the recent TabICLv2 model, using the PyPI package version 2.0.1. As opposed to the original TabICL variant, TabICLv2 natively supports both classification and regression. We use the default initialization parameters, in particular using 8 ensembled estimators for inference.

**AutoGluon:** We evaluate AutoGluon version 1.5 in the `extreme` preset with a time limit of 1h, fitted on a 40-core node with 320 GB of RAM and a single H100 GPU. We also evaluated the `best` preset with a time limit of 4h but found the former to be slightly better performing, likely due to the better use of recent tabular in-context learners in the model pool.

**PytabKit:** We follow the setup in ConTextTab, but use the recent 1.7.3 version of the PyPI `pytabkit` package (Holzmüller et al., 2024) with updated search spaces following the TabArena setup (Erickson et al., 2025). For RealMLP, we report the best-in-class tuned Caruana-ensembled results (Erickson et al., 2025). For the extended results, we also show the tuned-defaults (TD) variants.

**Scikit-learn:** For the random forest baseline, we use the corresponding scikit-learn (Pedregosa et al., 2011) estimators with their default settings. We use scikit-learn version 1.5.2.

### A.6. Evaluation metrics

For all metric calculations, we use the corresponding scikit-learn implementations (Pedregosa et al., 2011).

For classification and regression results, we report mean accuracy and soft-clipped $R^2$: due to the dominance of outliers, we soft-clip the negative $R^2$ values to the range $[-1, 0)$ via $\tanh$, which is smoothly differentiable and retains the relative ordering of negative scores (Spinaci et al., 2025).

For the calculation of the Elo scores, we utilize the implementation of TabArena (Erickson et al., 2025), which itself is based on ChatBot Arena's implementation (Chiang et al., 2024).

For all ranking results (aside from those used in the critical-difference diagrams), we use a robust ranking algorithm. That is, models that lie within 0.1 percentage points (in terms of either Accuracy, AUROC, or $R^2$) are considered ties. The same

robustification also applies to our Elo score and win-ratio calculations.

For the critical-difference (CD) diagrams, we use the `autorank` library (Herbold, 2020). So as not to influence the statistical guarantees of the significance test used, we do not alter or robustify its internal rank calculation. Hence, the ranks shown in the CD diagrams can deviate from those shown in the result tables or figures.

### A.7. Datasets

**Classification and regression:** We evaluate all models on CARTE, TextTab, TabArena-Lite, and TALENT-Tiny. We use the same split creation protocol as ConTextTab, except for TabArena-Lite, for which we use the predefined splits.

**Matching:** We evaluate on the synthetic Febrl4 dataset from the Freely Extensible Biomedical Record Linkage project (Christen, 2008), as well as on Fodors-Zagats, Bikes, eBooks, and Movies, sourced from the DeepMatcher study (Mudgal et al., 2018) as well as the Magellan Data Repository (Das et al.). Due to the relatively small size of these datasets with available ground truth match labels, we perform 5-fold cross-validation over all available labeled examples and report average match-class F1 scores.

## B. Additional Results

### B.1. Classification and regression

**Additional baseline evaluations:**

*Table 3.* Extended classification and regression performance, depicting mean accuracy (Acc) for classification and (soft-clipped) $R^2$ score for regression tasks, in percent, as well as mean per-task rank.

| Model | All | CARTE | | | TabArena-Lite | | | TALENT-Tiny | | | TextTab | | |
|---|---|---|---|---|---|---|---|---|---|---|---|---|---|
| | Rank | Rank | Acc | R2 | Rank | Acc | R2 | Rank | Acc | R2 | Rank | Acc | R2 |
| AutoGluon v1.5 | N/A | N/A | 78.8 | 73.7 | N/A | 88.1 | 79.9 | N/A | 87.8 | 85.3 | N/A | 83.5 | 67.5 |
| **FlexTab (ours)** | **2.7** | **1.5** | **77.3** | **74.4** | 3.5 | 87.1 | 79.5 | 3.1 | 88.0 | **85.8** | **2.8** | 83.8 | 62.7 |
| RealMLP | 3.2 | 3.4 | 74.3 | 69.0 | 3.0 | 88.3 | 79.9 | 3.1 | **88.6** | 85.0 | 3.2 | 82.3 | **67.7** |
| ConTextTab | 3.2 | 2.4 | 77.1 | 72.4 | 3.8 | 87.6 | 77.8 | 3.7 | 87.6 | 83.2 | 3.2 | **84.4** | 58.8 |
| TabPFN v2.6 | 3.3 | 5.4 | 70.7 | 59.9 | **1.8** | 88.7 | **80.5** | 2.2 | 88.4 | 84.9 | 3.9 | 81.5 | 63.8 |
| TabICL v2 | 3.4 | 5.6 | 70.5 | 55.3 | 2.0 | **88.7** | 79.9 | **2.1** | 88.3 | 85.3 | 3.6 | 82.5 | 60.2 |
| CatBoost | 3.4 | 3.5 | 76.3 | 68.3 | 3.5 | 88.2 | 79.0 | 3.7 | 87.1 | 83.5 | 2.9 | 83.7 | 65.4 |
| Random forest | 5.3 | 5.8 | 71.4 | 63.4 | 4.9 | 87.6 | 76.1 | 5.2 | 85.6 | 78.0 | 5.0 | 79.8 | 60.7 |

Additional results are shown in Table 3. The overall results are in line with the selected presentation in the main paper and serve only to illustrate the complete picture. Across all evaluated models and benchmarks, FlexTab ranks best and performs particularly well on semantic-heavy benchmarks. This is in line with the results of ConTextTab, which our architecture builds upon, while narrowing the gap on numeric-heavy regression, such as on TabArena-Lite, but remaining behind in classification.

**Relation between dataset size and model performance:** We plot the average rank of each model as a function of the dataset size expressed in the number of rows and the number of columns in Figure 5 across all evaluated datasets.

We observe that FlexTab performs particularly well in the 1 k to 10 k row count regime. For smaller tables, TabICL and TabPFN tend to perform better, but we also note the large confidence intervals due to the comparatively small support set in those areas. For larger tables, the in-context learners tend to converge to a similar performance and start being outperformed by CatBoost and RealMLP, which are extensively trained and tuned for each dataset.

In terms of the dependence on the number of columns or features of each task, we observe that FlexTab excels at narrow tables, but shows decreasing performance for wider ones (relative to the other models). Note again the potentially small support set and wide confidence bands, and also that a ranking score can exaggerate such differences and that the absolute differences may not be as large. Nevertheless, this points to a potential bottleneck in our architecture: As we compress each row into a single embedding that is fed to the prediction decoder, wider tables naturally undergo a relatively larger compression than narrow ones. This should be investigated in future work, for example by increasing the hidden dimension or investigating hybrid approaches that utilize multiple tokens per row for wider tables.

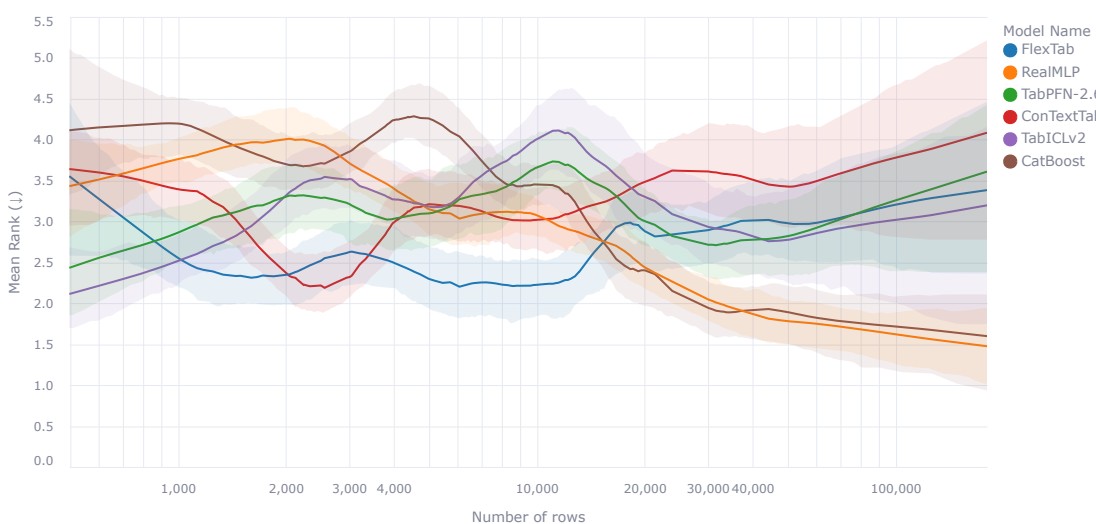

*(a)* Performance dependence on row count.

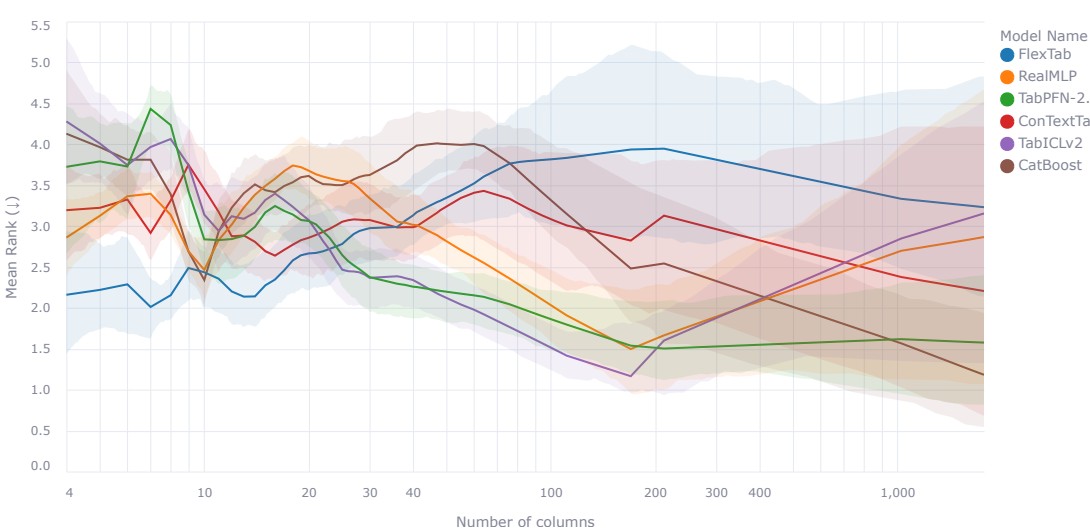

*(b)* Performance dependence on column count.

*Figure 5.* Relation between number of training dataset rows (top) and columns (bottom) and performance, obtained as a LOWESS regression in the plane $\log(n_{\text{rows/cols}}, \text{rank})$. The confidence bands are the 80% confidence intervals obtained via bootstrapping.

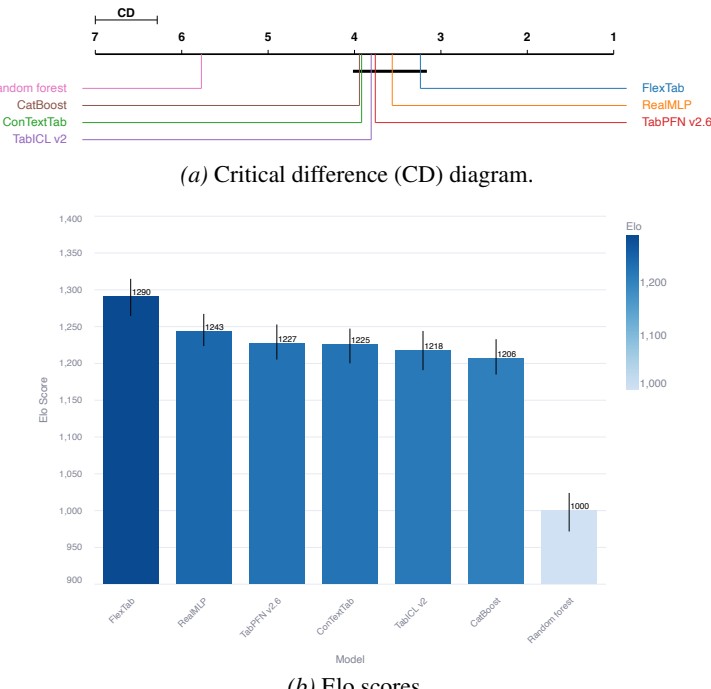

*(a)* Critical difference (CD) diagram.

*(b)* Elo scores.

*Figure 6.* Critical difference (CD) diagram (top) and Elo scores (bottom) of the main investigated models across all 158 datasets from CARTE, TabArena-Lite, TALENT-Tiny, and TextTab benchmarks.

**CD diagrams and Elo scores:** Additionally, we show CD diagrams and Elo scores across all investigated benchmarks (CARTE, TabArena-Lite, TALENT-Tiny, and TextTab) in Figure 6. Wins are calculated based on accuracy for classification and $R^2$ for regression datasets.

For fine-grained insights, we also plot per-benchmark-type diagrams:

- for semantic-heavy benchmarks (CARTE and TextTab) in Figure 7

- for numeric-heavy benchmarks (TabArena-Lite and TALENT-Tiny) in Figure 8.

Across all investigated benchmarks, FlexTab shows very strong performance, outperforming other baselines, even hyperparameter-optimized ones with inner CV ensembles. FlexTab achieves the highest Elo score across the full set of 158 datasets (Figure 6).

The per-benchmark-type analysis reveals a more nuanced picture. On the 71 semantic-heavy datasets from CARTE and TextTab (Figure 7), FlexTab's advantage is particularly pronounced; it reaches an Elo of 1498, more than 100 points ahead of the next-best in-context learner (ConTextTab) and more than 400 points ahead of TabPFN v2.6 and TabICL v2, whose handling of semantics is more limited. The corresponding critical-difference diagram shows that FlexTab significantly outperforms all models apart from ConTextTab. On the 87 numeric-heavy datasets from TabArena-Lite and TALENT-Tiny (Figure 8), the picture is tighter: TabPFN v2.6 and TabICL v2 take the top two Elo positions, with FlexTab ranking fourth but remaining competitive with all tuned and ensembled per-dataset baselines. Notably, the critical-difference diagram indicates that the top three models in this regime are statistically indistinguishable.

These results show that FlexTab is the strongest method overall across the investigated benchmarks, with a clear and statistically significant edge on semantic-heavy tasks and competitive performance on numeric-heavy ones. This supports our central claim that target-agnostic row embeddings, combined with task-specific decoders, provide an effective and broadly applicable foundation without sacrificing performance on the well-studied classification and regression regimes.

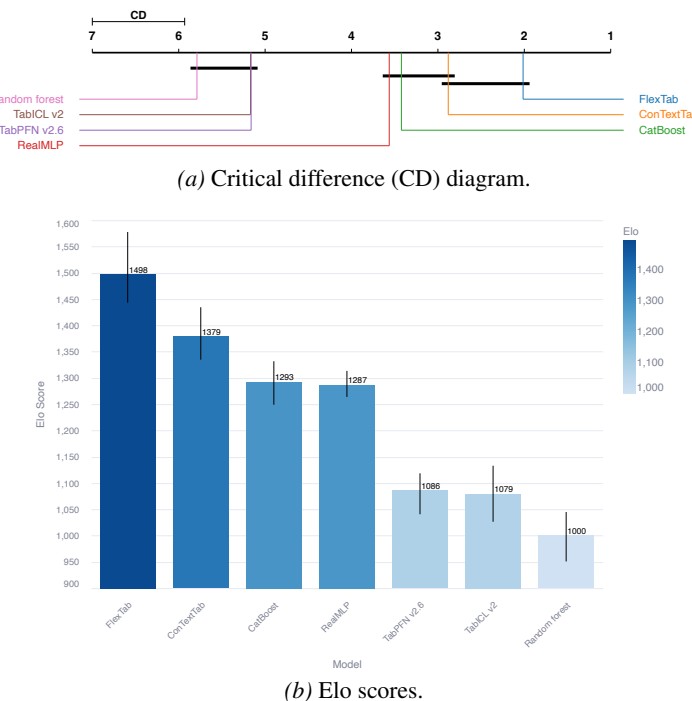

Figure 7. Critical difference (CD) diagram (top) and Elo scores (bottom) across 71 semantic-heavy datasets from CARTE and TextTab benchmarks.

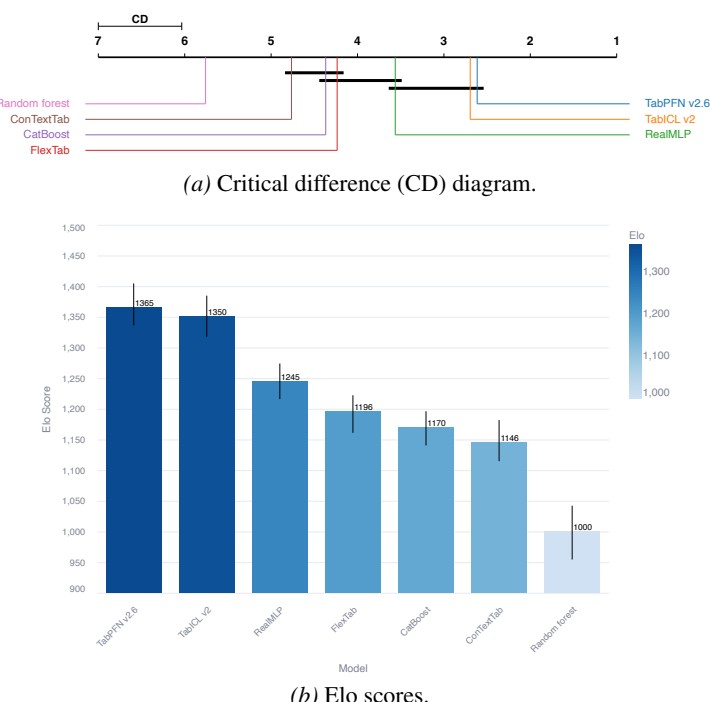

Figure 8. Critical difference (CD) diagram (top) and Elo scores (bottom) across 87 numeric-heavy datasets from TabArena-Lite and TALENT-Tiny benchmarks.

*Table 4.* Matching benchmark F1 scores (%, positive class) obtained as 5-fold cross-validation averages for each dataset. The average per-group rank is denoted as GRk whereas the average rank across all investigated models is denoted as Rk.

| Model | Benchmark | | | | | Average | | |
|---|---|---|---|---|---|---|---|---|
| | Febrl4 | F.-Zagats | Bikes | eBooks | Movies | F1 | GRk | Rk |
| *In-context learners* | | | | | | | | |
| TabPFN | 97.3 | 29.4 | 81.8 | 67.9 | 0.0 | 55.3 | 3.6 | 10.0 |
| TabPFN (TF-IDF) | 94.8 | 0.0 | 79.6 | 67.7 | 31.2 | 54.7 | 5.0 | 11.0 |
| TabICL | 95.0 | 36.9 | 82.4 | 73.1 | 0.0 | 57.5 | 3.2 | 9.6 |
| TabICL (TF-IDF) | 76.0 | 4.7 | 82.4 | 67.4 | 5.9 | 47.3 | 5.0 | 11.4 |
| ConTextTab | 83.2 | 23.2 | 82.3 | 60.3 | 79.0 | 65.6 | 4.2 | 9.6 |
| **FlexTab (ours)** | 79.8 | 12.7 | 84.3 | 72.6 | 61.0 | 62.1 | 3.8 | 9.4 |
| **FlexTab-Match (ours)** | **99.3** | **92.5** | **86.8** | **87.3** | **90.2** | **91.2** | **1.0** | **2.4** |
| *Embedding similarity* | | | | | | | | |
| NGramHash Embeddings | **99.7** | **88.0** | 69.8 | **88.1** | 22.9 | 73.7 | **1.6** | **6.4** |
| Skrub Embeddings | 45.0 | 61.4 | 47.6 | 53.1 | 31.4 | 47.7 | 3.4 | 12.6 |
| TabPFN Embeddings | 71.8 | 41.2 | 50.8 | 56.4 | 24.0 | 48.9 | 3.2 | 12.6 |
| **FlexTab Embeddings (ours)** | 98.5 | 81.5 | 54.4 | 84.9 | **78.9** | **79.6** | 1.8 | 7.8 |
| *Per-dataset trained* | | | | | | | | |
| Deepmatch SIF | 99.5 | 0.0 | 0.0 | 35.6 | 0.0 | 27.0 | 4.2 | 12.4 |
| Deepmatch RNN | 99.9 | 89.5 | 86.0 | 88.1 | 80.2 | 88.8 | 2.2 | 2.8 |
| Deepmatch Attn | 99.9 | 89.5 | 82.1 | 84.0 | 83.4 | 87.8 | 2.6 | 4.0 |
| Deepmatch Hybrid | 99.8 | 87.1 | 87.7 | 88.5 | **87.6** | **90.2** | 2.0 | 2.6 |
| DITTO | **100.0** | **94.7** | **95.5** | **89.5** | 44.9 | 84.9 | **1.6** | **2.4** |

## B.2. Matching

We provide additional results in Table 4. These include TabPFN and TabICL combined with TF-IDF features obtained for high-cardinal columns via Skrub's `TableVectorizer`, as well as the complete model list of the DeepMatcher evaluation (Mudgal et al., 2018). Finally, we also report results using embedding-based matching, using TabPFN embeddings as well as FlexTab's native target-agnostic embeddings. The prediction is obtained from the pairwise cosine similarities of the two embeddings, with a threshold tuned on each train fold of each dataset.

Overall, the results are in line with the distilled ones presented in the main paper. The matching-native FlexTab-Match performs best across all models considered – in terms of both average F1 and rank. Moreover, we observe that the embeddings obtained from our encoder clearly outperform those from TabPFN when used for matching, highlighting the benefit of the general-purpose, target-agnostic row embeddings produced by our architecture. Nevertheless, for many matching tasks, simple n-gram hash embeddings perform very well, while their performance strongly degrades on others (e.g., Bikes and Movies). Overall, we can already achieve reasonable results with simple cosine-similarity-based matching using FlexTab's row embeddings, which can be useful in large-scale matching scenarios. However, to achieve the best performance, the native matching decoder is preferred.

