# OpenReview forum: "FlexTab: Towards a Flexible Encoder-Decoder Architecture for Tabular In-Context Learning"
_ICML.cc/2026/Workshop/FMSD — FMSD @ ICML 2026 Poster_

### Official Review · Reviewer_Ecnj · 2026-05-16
**Strong and relevant flexible architecture for tabular in-context learning**

**Rating:** 7
**Confidence:** 3

**Review:**

**Summary:**

The paper proposes **FlexTab**, an encoder-decoder architecture that separates a shared target-agnostic row encoder from task-specific decoders for classification, regression, and entity matching. This is a useful direction.

**Strengths:**

The paper is well aligned with the workshop, technically clear, and addresses an important limitation of existing tabular foundation models. The results are strong across classification/regression benchmarks and especially promising for semantic-heavy datasets and entity matching. The matching decoder is a good demonstration that the architecture can go beyond standard single-table prediction.

**Weaknesses:**

 The flexibility claim is only partially demonstrated, since only one clearly non-standard task, entity matching, is evaluated. The proprietary pretraining corpus and limited contamination analysis also weaken confidence in the benchmark results. More ablations are needed to isolate the contribution of the target-agnostic encoder, task-specific decoders, and layer aggregation. Runtime and training cost should also be discussed more clearly.

**Detailed comments:**

The matching setup should be better positioned relative to full entity matching pipelines, which usually include blocking and candidate generation. The related work could also mention recent embedding-based table representation methods such as TabEmb and TabEmbed.

**Justification:**

 This is a strong and relevant workshop paper with a meaningful architectural contribution and convincing initial results. I recommend acceptance, while encouraging stronger ablations and contamination analysis.

---

### Official Review · Reviewer_p7Ad · 2026-05-22
**Interesting idea; but the worth is not that well proved**

**Rating:** 6
**Confidence:** 4

**Review:**

The paper proposes FlexTab, a flexible encoder-decoder architecture for tabular in-context learning. The main idea is to separate target-agnostic row embeddings from task-specific decoders, allowing the same encoder to support classification, regression, and entity matching tasks.

The paper has its strengths strong points as the idea of decoupling the encoder-decoder is interesting and the results on semantic benchmarks are strong. The paper is also fairly well written and the experiments are broad enough to show the flexibility of the method.

However, there are still some weaknesses.
1. The novelty feels somewhat incremental since the work mainly extends existing tabular transformer architectures with a more modular setup.
2. The evaluation also relies heavily on data that is not publicly released, so claims are not that easy to review.

In addition, the method seems to struggle on wider tables and numeric-heavy datasets compared to some baselines, so does that mean that the proposed row compression is not fit for large datasets? Similar trend is for matching experiments as they are on a small benchmark, so it is unclear how well the approach would generalize to larger real-world settings.

---

### Official Review · Reviewer_BK7b · 2026-05-22

**Rating:** 6
**Confidence:** 3

**Review:**

The paper proposes FlexTab with a target-agnostic encoder and task-specific decoders. The encoder creates reusable row embeddings, while different decoders handle classification, regression, and entity matching. The goal is to avoid tying representations to one target column. Overall, this is a decent workshop paper. The architecture is technically sound, but the evidence is not enough to prove FlexTab is a fully general-purpose tabular foundation model.

Strengths : The encoder-decoder split is a clean idea. It makes FlexTab more flexible than other models. The paper tests more than standard classification/regression. The entity matching decoder is interesting, and FlexTab-Match gets strong results against DeepMatcher and DITTO on the selected datasets. The model is competitive on classification/regression, especially on semantic-heavy benchmarks.

Areas for Improvement : The “general-purpose” claim is too strong. The paper tests classification, regression, and matching, not clustering, outlier detection, imputation, relational prediction, or temporal tasks. Data contamination is a concern. The model is pretrained on around 300k real-world unlabeled tables, but the paper does not run a contamination study against the evaluation benchmarks.

Detailed Comments : The authors should clarify the main goal: is FlexTab a stronger single-table predictor, a reusable representation model, or a general multi-task tabular foundation model? The evidence best supports the reusable representation model framing. The single-row embedding may be a bottleneck. Compressing each row into one vector may hurt wide tables or tasks that need detailed cell-level reasoning. The appendix also shows weaker behavior on wider tables.  More ablations would help: frozen vs jointly trained encoder for matching, one row token vs multiple row tokens, decoder depth, shared vs separate decoder weights, and transfer to another task like imputation or anomaly detection. The matching result is promising, but the setup is simplified. The paper evaluates only the prediction step, not the full entity matching pipeline with blocking or candidate generation.

Justification of Score : The paper is technically sound and has a useful architectural idea, but it is not yet fully convincing as a general-purpose tabular foundation model. The main positives are the clean encoder-decoder design, good semantic-table results, and strong matching decoder. The main concerns are limited task coverage, contamination risk, simplified matching setup, and limited validation beyond the chosen benchmarks.